# The Integrated Family Approach in Mental Health Care Services: A Study of Risk Factors

**DOI:** 10.3390/ijerph21050640

**Published:** 2024-05-17

**Authors:** Hanna Stolper, Marjolein van der Vegt, Karin van Doesum, Majone Steketee

**Affiliations:** 1Department of Psychology Education and Child Studies, Erasmus University Rotterdam (EUR), 3062 PA Rotterdam, The Netherlands; majonesteketee@verwey-jonker.nl; 2Jeugd ggz Dimencegroep, 8017 CA Zwolle, The Netherlands; m.vandervegt@jeugdggz.com; 3Department of Clinical Psychology, Radboud University Nijmegen, 6525 XZ Nijmegen, The Netherlands; karin.vandoesum@ru.nl; 4Impluz Dimencegroep, 7411 GT Deventer, The Netherlands; 5Verwey-Jonker Instituut, 3522 KE Utrecht, The Netherlands

**Keywords:** integrated family approach, family-focused practice, parental mental disorders, infancy and early childhood, adverse child experiences (ACE), case file study, risk factors

## Abstract

Background: Parental mental disorders in families are frequently accompanied with other problems. These include family life, the development of children, and the social and economic environment. Mental health services often focus treatments on the individual being referred, with little attention to parenting, the family, child development, and environmental factors. This is despite the fact that there is substantial evidence to suggest that the children of these parents are at increased risk of developing a mental disorder throughout the course of their lives. Young children are particularly vulnerable to environmental influences given the level of dependency in this stage of development. Objective: The main objective of this study was to identify whether there were a complexity of problems and risks in a clinical sample of patients and their young children (0–6) in mental health care, and, if so, whether this complexity was reflected in the integrated treatment given. Methods: The data were collected for 26 risk factors, based on the literature, and then subdivided into the parental, child, family, and environmental domains. The data were obtained from the electronic case files of 100 patients at an adult mental health service and the corresponding 100 electronic case files of their infants at a child mental health service. Results: The findings evidenced a notable accumulation of risk factors within families, with a mean number of 8.43 (SD 3.2) risk factors. Almost all of the families had at least four risk factors, more than half of them had between six and ten risk factors, and a quarter of them had between eleven and sixteen risk factors. Furthermore, two-thirds of the families had at least one risk factor in each of the four domains. More than half of the families received support from at least two organizations in addition to the involvement of adult and child mental health services, which is also an indication of the presence of cumulative problems. Conclusion: This study of a clinical sample shows clearly that the mental disorder among most of the patients, who were all parents of young children, was only one of the problems they had to deal with. The cumulation of risk factors—especially in the family domain—increased the risk of the intergenerational transmission of mental disorders. To prevent these parents and their young children being caught up in this intergenerational cycle, a broad assessment is needed. In addition, malleable risk factors should be addressed in treatment and in close collaboration with other services.

## 1. Introduction

Epidemiological research shows that the children of parents with a mental disorder are at serious risk of developing mental disorders themselves [1,2]. For infants and young children, this risk is significantly higher because early childhood is a critically vulnerable period for brain development and a period of high sensitivity to stress. The exposure to stress early in life is associated with later psychopathology, for instance, anxiety, depression, ADHD, and post-traumatic stress disorder [3,4]. In general, the transmission of mental disorders varies and depends on the number of risk and protective factors [5]. Several studies indicate that adversities in childhood often co-occur [6], and these studies find a dose-dependent response of these adversities to mental health outcomes [7,8,9]. Thus, the impact of children exposed to multiple risk factors is greater than if they are exposed to a single risk factor. For instance, Barker et al. (2012) [8] found that, for a two-year-old child in the presence of a mother with a depression, each additional risk factor increases the odds of the child developing a mental disorder by at least 20% at the age of seven.

The explanation for this relationship is assumed to be that risk factors in different domains—parent, child, family, environment—will negatively interfere with parental functioning [10], and, hence, the quality of the parent–child relationship is compromised. For young children, the quality of parent–child interactions is critical for the development of a secure attachment, which, in turn, is a protective factor for the development of mental disorders [11,12,13]. Thus, in this paper, we present our research regarding the prevalence of risk factors, and the relationships between certain risk factors and the domains in which they occur in a clinical population of patients with mental disorders who are also parents of young children.

In our review [14], we have presented a summary of risk and protective factors in the process of the intergenerational transmission of mental disorders in early childhood, and we have subsequently formulated intervention targets of how to support parents of infants and young children in mental health care. Risk and protective factors were identified in different domains, namely the parental, family, child, and environmental domains. These interacting domains affect the developing early parent–child relationship (see Figure 1).

Because of the interrelatedness of the domains, there might be positive or negative cascading effects depending on the interaction between the present risk and the protective factors. For instance, (parental) mental disorders and impaired social and economic functioning often go together and cause each other [15], which then aggravates the burden. Likewise, in a huge epidemiological study [6], it was found that childhood adversities associated with risk factors in the parental domain, such as parental mental disorders, and in the family domain, such as interparental violence, criminal behavior, neglect, and physical and sexual abuse of the child, were the strongest predictors of mental disorders over the course of the child’s lifetime.

According to the cumulative risk model [16], several studies indicate a dose-dependent response relation of childhood adversities and trauma to mental health outcomes, which means that the number of traumas and adversities is a significant predictor of mental health disorders [3,8,9,17,18,19,20,21]. Additionally, the adverse child experiences (ACEs) study [22] has demonstrated that traumas in childhood up to the age of 18 can lead to medical diseases and mental disorders. Furthermore, several studies have shown that the risk is significantly increased in the presence of four or more ACEs. For young children of up to six years of age, it was found that children with three or more ACEs, compared with children with no ACEs, were significantly more likely to exhibit behavioral problems (e.g., aggression, attention problems), mental health problems (e.g., anxiety), and overall problems [21].

The original three categories of ACEs, which included ten empirically selected ACEs, are mostly related to family functioning, abuse (emotional, physical, sexual), neglect (physical, emotional), and household dysfunction (parental violence, addiction, imprisonment, a parent with a mental disorder or in a psychiatric hospital, or not being raised by both biological parents) [21]. Interestingly, there is some evidence of an association of parental ACEs and child ACEs, which suggests an intergenerational pathway of transmission [23]. For instance, parents with ACEs may develop symptoms of post-traumatic stress disorder (PTSD), which is recognized as a risk factor for difficulties in the parent–child relationship, negative parenting practices, and ACEs in their children [23]. A notable list of the early consequences of ACEs in children were mentioned [24], such as attachment disorders, behavioral problems, anxiety and depression, and suicide attempts. However, the variability outcomes illustrate that this is not a deterministic process, and there is evidence that resilience factors and processes may provide protection against the impact of ACEs [25].

Other researchers suggest that it is not only the presence of the parental mental disorder and its consequences on parenting which determine the child’s outcome, but that the presence of other risk factors are independently or partly responsible [8,26]. The ACE study found that the impact of the different ACEs, of which a parent with a mental disorder is one, is more or less the same, although sexual abuse is found to be the most synergistically reactive ACE [27]. A particular combination that results in high risk for poor child outcomes is the presence of a parental mental disorder and poverty [28]. Another study found that the children of parents with a mental disorder combined with low economic status are more affected than the children of a higher socio economic status [29]. The aforementioned study of Barker et al. [8] revealed that the children of parents with a mental disorder, when compared to the children of parents without a mental disorder were, on average, exposed to, respectively, 2.3 risk factors versus 1 risk factor. These 10 risk factors are a low socioeconomic status, a single caregiver, partner cruelty, low partner affection, low emotional and low practical support networks, early parenthood, low educational attainment, substance use, and criminal trouble with police.

Referring to the abovementioned research, the impact on young children exposed to multiple risk factors is greater than if they are exposed to a single risk factor [6,7,9], and there is some evidence that, in the presence of multiple risk factors, the parental mental disorder has a priori no more impact than other risk factors. In consideration of this, it would be interesting to investigate whether there are actually multiple risk factors to be identified in the population of parents with a mental disorder who have young children, apart from the parent’s mental disorder and, in that case, the number of these present risk factors, as well as in how many domains they are present (parent, child, family, environment).

In this paper, we will present our findings of an exploratory case file study of parents and their young children, both referred for an integrated family approach in a treatment conducted by professionals of an adult and a child mental health care service. In the Netherlands, an integrated family approach was conducted in mental health care, which “involves multi-disciplinary treatment for the family as a whole. Treatment using this approach offers parents with a diversity of mental disorders and their young children a combined treatment addressing current problems in different domains within the family, namely the mental disorder(s) of the parent(s), the partner relationship, parenthood and family life, the parent–child relationship, and the child’s mental or relational disorder. The aim of this collaborative integrated family approach is to increase the quality and efficiency of the treatment for parents and their young children, to improve their relationships, and to ameliorate the risk of intergenerational transmission of psychopathology or other adverse outcomes” [30,31].

The aim of this current case file study is to investigate whether, in this clinical sample, there are actual problems in different domains, and if any associations can be found between the number of risk factors. Furthermore, we will evaluate if the domains in which the risk factors occurred are addressed in treatment, as was intended by the integrated family approach. In addition, we will support professionals in clinical practice in identifying risks in families with a parent suffering from mental disorders, as recommended as standard practice [32,33].

### Current Study

The research questions of the current study were as follows: first, what is the prevalence of each of the 26 risk factors in this sample, and what is the variation of these risk factors across the four domains, which are parent, child, family, and environment? Because of the assumed interrelatedness between the different domains, we expect the majority of the families to have problems in at least three of the domains, namely parent, child, and family, and a number of them will experience problems in the environment. Second, are there any correlations between the domains in which the risk factors are found? We expect stronger associations between the parental, child, and family domains than between these three domains and the environmental domain. Third, is there a correlation between the four risk factors associated with the severity of the parental mental disorder, namely the age at the time of the first onset, chronicity, the comorbidity of the mental disorder, and the number of ACEs. We expect an association between the number of experienced traumas in parental childhood and the other three parental risk factors. Fourth, is there a correlation between the number of parental ACEs and the number of the child’s ACEs? Although previous research has shown a correlation between parents’ and children’s ACEs, we expect no correlation. Given the young age of the children in this study, and the fact that the final ACE score is based on children of 0–18 years of age, we consider it unlikely that an association would be seen before the age of six. Fifth, is the level of identified risk factors (in different domains) reflected in the intensity of the treatment? Consistent with the aim of an integrated family approach in treatment, we expect that the number of identified risk factors is reflected in the intensity of the treatment.

By finding answers to these research questions, we aim to provide mental health professionals with knowledge that allows them to tailor their treatment to support parents in preventing the transmission of problems to their children.

## 2. Materials and Methods

### 2.1. Sample

The present (retrospective) case file study was conducted at an adult mental health service (AMHS) and with infant mental health teams at a child and adolescent mental health service (CAMHS). Both services are part of the Dimence Groep, a mental health care foundation in medium-sized cities in the northeast part of the Netherlands. Both outpatient services of adult and child mental health care offer specialized multidisciplinary mental health care.

Ethics approval was granted by the Medical Ethics Review Board at the University Medical Centre of Utrecht in the Netherlands (18–186/C). All included adult patients in the period of 2013–2023 provided permission for the retrieval of any anonymous data from their electronic files that was relevant to the study. For the case files of the child, permission was required from the parents who had custody of the child. In this study, the electronic case files of 100 patients and their 100 young children aged 0–6 years were analyzed. In all treatments, an integrated family approach was used, in which the professionals from AMHS and CAMHS worked collaboratively, tailoring the various treatment components to the family’s capabilities and particular context. The case files are administrated by involved professionals.

### 2.2. Participants and Recruitment

Because we started the entire study in 2018, families of two groups were selected for participation, namely families who were still in treatment (period after 2018) and families in which the treatment had been completed (2013–2018). The two groups were equal. Parents of the first group were informed and asked for participation by their own therapist. Of this group, 74 families consented. At the time of the data analysis, five families were still undergoing treatment. Thirty-six families of this first group refused, mostly due to the burden of their situations. The second group, in which treatment had already concluded, was approached via telephone or email, or, if possible, by their own therapist in order to improve the chance of consent for participation. In case the therapist was unable to do this or was no longer in service, someone from the research team contacted them. Twenty-six families consented, and twelve refused consent for various reasons. The most frequent reason for not participating in the study was that they did not want to be reminded of the difficult period of their lives when they were in treatment. For a few parents, the reason was the treatment itself, of which they did not want to be reminded. Two parents we could not reach because their contact details were out of date.

### 2.3. Procedure, Data Collection

The data were collected through analyzing the case files of the parent(s) and the child receiving the treatment. A format with relevant variables for the patient/parent, child, family, and environment was utilized for this data collection. Demographic and treatment variables were collected for the parent and the child. The following treatment variables were examined: the DSM-5 classifications, number of clinical interventions, duration of treatment, involved disciplines, and involvement of other services. Data were collected and discussed by two researchers (MvdV, general psychologist, HS, clinical psychologist, PhD candidate). If the information from the case file was not clear, the therapist of the parent or child was consulted if possible, and when the lack of clarity continued to exist, the variable was characterized as missing data. The data were collected and transferred to SPSS.

The classifications of the mental disorders of the parent and the infant, derived from the case files, were based on multidisciplinary examination involving a psychologist, psychotherapist, and psychiatrist, who are all authorized and qualified accordingly. Regarding the infants, these professionals were additionally educated in infant mental health and early childhood (IMHEC).

### 2.4. The Participants

A total of 100 parents and their young children were included. The parents’ and children’s characteristics at the time of referral are shown in Table 1. More mothers than fathers participated in the study. The percentages of the genders of the participating children were almost equal. Almost half of the children were younger than one year old (46%), and far more than two-thirds (72%) were under three years old. The highest education attainment achieved by half of the parents (50%) was of a medium level, 20% achieved a low level, and 30% achieved a high level of education. This is a reasonably realistic reflection of the population in the society of the Netherlands. When compared to the latter, in our sample, the middle level of education was slightly higher, at the expense of the upper and lower levels. 

There were no exclusion criteria concerning the assigned DSM-5 classifications, either for the parents or the infants. For instance, adult patients with the following classifications were included: personality disorder, bipolar disorder, depressive disorder, anxiety disorder, autism spectrum disorder, post-traumatic disorder, and other specified trauma and stressor-related disorders. Regarding the included infants, the following classifications were diagnosed: autism spectrum disorder, unspecified neurodevelopmental disorder, post-traumatic stress disorder, and parent–child relational problems.

### 2.5. Measurements

Measurements were based on a previous study of the risk factors associated with the intergenerational transmission of mental disorders in the four following domains: parent, child, family, and environment [14].

Individual adult/parental risk factors: the early onset of the mental disorder, measured using the age of onset before age 30 [5]; comorbidity, measured using more than one DSM-5 classification; chronicity, measured via ≥ 2 referrals to mental health care services in their history; adverse childhood experiences (ACEs), measured via the number of ACEs in the period of 0–18 years. From the original ACEs framework, we used three categories which included ten empirically selected ACEs as follows: abuse (emotional, physical, sexual), neglect (physical, emotional), and household dysfunction (parental violence, addiction, imprisonment, mentally ill or in psychiatric hospital, not being raised by both biological parents) [21]. An individual ACE score was calculated via counting the number of categories experienced in childhood (0–10). Epidemiological research showed that the presence of ≥ 4 ACEs is a risk factor for physical and mental health problems [22].

Child risk factors: mother had mental health problems during pregnancy [4]; comorbidity, measured via more than one DSM-5 classification; chronicity, measured using the number of previous child mental health or youth care services; number of ACEs (0–10). The presence of ≥ 3 ACEs is confirmed to be significant risk factor for young children (age 0–6) to exhibit behavioral problems (e.g., aggression, attention problems), mental health problems (e.g., anxiety), and overall problems when compared with children with no ACEs [21].

Family risk factors. Single parenthood, no co-parent or shared care with other biological parent; early parenthood, aged below 20 years [8,10]; both parents experienced mental disorders, measured via the treatment period of the other parent at a mental health care service; family member suffers with addiction; parents divorced; problems in the couples’ relationship, as mentioned in the case file or addressed in couples therapy or youth care; domestic violence, this includes partner violence, physical/sexual/emotional abuse, and neglect, and was measured using the notes in the case file and/or the involvement of child protection services; other child with problems, measured using the notes or involvement of other child care service; life events, measured via the physical illness of a family member, birth of a child, moving, or the death of a family member; low education attainment, measured via a basic or pre-vocational secondary education; child-rearing problems, unpredictable routines, or a lack of daily routines, measured via the involvement of home treatment or support at home.

Environmental risk factors. This category largely concerns socio-economic factors, as follows: unemployment, missing work, or problems at work; financial problems, addressed by social services, church, or extended family; housing problems, measured using notes about a house that is too small in relation to the family size, due to divorce or financial strains; criminality, notes about trouble with the police or imprisonment; problems within the extended family, measured via no contact or conflict with (grand)parents or a family member; no supportive social network or isolation, measured using notes in the case file; many professional services involved, measured via the number of services involved beside AMHS and CAMHS, with ≥2 other services involved being measured as a risk factor.

Treatment factors. The duration of the integrated treatment approach used by AMHS and CAMHS, measured in months; the number of interventions at AMHS and CAMHS (e.g., psychotherapy, pharmacotherapy, home treatment); the number of involved disciplines of AMHS and CAMHS (e.g., psychiatrist, psychologist, nurse practitioner); and the number of involved services besides AMHS and CAMHS (e.g., child protection, social services).

### 2.6. Data Analysis

All analyses were performed using SPSS version 27. To answer our first research question, descriptive frequency analyses were used for the whole sample to describe the prevalence of each of the 26 risk factors in this sample, alongside the variation of these risk factors across the four domains (parent, child, family, and environment). To answer our second research question, we converted Pearson’s r to ascertain if there are any correlations between the domains in which the risk factors were found. To answer our third research question of whether there was an association between the four parental risk factors regarding the severity of the mental disorder, we used Pearson’s r. To answer our fourth research question on whether there is a correlation between the number of parental ACEs and the number of the child’s ACEs, we also used Pearson’s r. Finally, for our last research question about the correlation between the intensity of the treatment and the level of identified risk factors in different domains, we used Pearson’s r. Although the assumption of normality required for using Pearson’s r was only not met for our fifth research question, the large sample size (N = 100) justifies its use.

## 3. Results

### 3.1. Frequencies of Risk Factors in Different Domains

Based on the investigations into the prevalence of the 26 risk factors in the sample population, our findings indicate a notable accumulation of these risk factors within the families, as detailed in Table 2, Table 3 and Table 4. The mean number of risk factors was 8.43 within the family (SD 3.2). It is evident that almost all of the families had at least four risk factors (96%). Far more than half of all families had 6–10 risk factors (56%), followed by a quarter of the families with 11–17 risk factors (25%). Furthermore, two-thirds of the families had at least one risk factor in each domain (66%), followed by a quarter of the families with at least one risk factor in three of the domains (25%).

Most families also have multiple risk factors within the different domains. Comparing the presence of risk factors in the four domains shows that the majority of problems that occur are connected to the severity of the mental disorder of the adult patients (parental domain). The data show that among almost all of them, multiple risk factors were found (85%). In the child domain, in which four risk factors were included, we found two or more risk factors in four out of ten families (41%). In the family domain, including 11 risk factors, we found four or more of these family risk factors in almost a quarter of the families (24%), and, in almost half of them (46%), we found two or three risk factors. In the environmental domain, including seven risk factors, we found four or more risk factors in one fifth of the families (21%) and two or three risk factors in four of the ten families (42%).

The most frequently reported risk factor in the domain of the parent (individual adult patient) was the early onset of the mental disorder. In nine out of ten patients, the mental disorder manifested before the age of 30. The age of the first onset of a mental disorder varied from 3 to 37, with a mean of 20.9 (SD 7.7) (Table 4). More than two-thirds of them were suffering with a comorbidity of mental disorders. The mean number of DSM-5 classifications was 2.2 (SD 1.1). More than half of them were suffering with chronic mental disorders, given the number of previous mental health treatments, with a mean of 3.1 (SD 1.8).

The most frequently reported risk factor in the child domain was the mothers’ mental health problems during pregnancy (N = 85 mothers, 82%), followed by children with three or more ACEs, making up almost one third of the sample (29%). Comorbidity was found in more than one fifth of the children, and a few children received previous care because of their problems.

In the family domain, more than half of the caregivers (56%) encountered difficulties with child upbringing and sought assistance to alleviate the situation. In more than one-third of the families (39%), there was a life event during treatment, such as the physical illness of a family member, the birth of a child, or relocation. Regarding the risk factors related to the family structure, we found that, in more than a quarter of the families, the parents were divorced, and, in one fifth of the families, there was a single parent. In almost one fifth of the families, one of the caregivers suffered from an addiction, and, in eight of the families, the other parent was also treated at a mental health service. Furthermore, in eight of the families, there were problems with one of the other children.

In the environmental domain, the majority of parents in treatment were unemployed or had experienced problems at work (67%), and, in one-third of the families (33%), there were financial problems. In far more than half of the families, there were at least two other services involved in addition to adult and child mental health services (57%), which is an indication of the complexity of the problems in these families. Regarding the risk factors related to social support, in 37 of the 100 investigated families, there were problems with the extended family, and, in seven families, there was no social support.

### 3.2. Frequencies of Adverse Child Experiences of the Parent and the Child

Most of the parents had experienced a cumulation of traumas in their childhood (Table 4 and Table 5). The mean of these ACEs was 2.8 (SD 2.2). More than one-third of all parents had experienced four or more of these childhood trauma’s (37%) (see Table 2), which was found to be associated with an increased risk of physical and mental health problems during their lifetime. Emotional neglect was reported by almost three of four parents, and almost half of the parents were not raised by both biological parents. Furthermore, 41% of the parents were exposed to a parental mental disorder as a child, and more than a quarter experienced physical, emotional, and sexual abuse.

Of the children, 29% had at least three ACEs, which was found to be associated with behavioral, mental health, and overall problems [21]. The mean of these ACEs was 2.08 (SD 1.43). Because one of the common characteristics of this sample was to have a parent treated at an adult mental health service, all of the children scored on this ACE, followed by 29% of the children who did not live with both biological parents. In addition, 14% experienced violence between the parents, and 26% experienced emotional neglect. Regarding abuse, eight children experienced physical and emotional abuse, and one child experienced sexual abuse.

Table 5 shows all of the 10 ACEs reported in this sample, both for parent and child, and Table 6 shows the mean ACEs and SD.

### 3.3. Correlations between the Parental, Child, Family, and Environmental Domains

There are correlations between the risk factors in different domains, but not all the risk factors of the domains correlated with each other. As can be seen in Table 7, a significant moderate correlation was found between the number of risk factors in the parental domain and the number of risk factors in the family domain (r = 0.31). Significant but weaker correlations were found between the risk factors of the child domain and the parental domain (r = 0.27), the child domain and the family domain (r = 0.24), and the family domain and the environmental domain (r = 0.23).

### 3.4. Significant Correlations between Risk Factors within the Parental Domain

As shown in Table 8, within the parental domain, we found a negative significant moderate correlation (r = −0.36) between the number of ACEs and the parent’s age at the first onset of a mental disorder. Thus, the higher the number of ACEs, the younger the age at which the mental disorder manifested for the first time, with a mean of age 20.7 (SD 7.7). Another negative significant correlation (r = −0.31) was found between the age at which the mental disorder manifested for the first time and the chronicity of the mental disorder. This result means that the younger the age of the onset of the mental disorder, the more likely it is that mental problems will tend to persist. A third significant moderate correlation (r = 0.32) was found between the chronicity of the mental disorder and the number of comorbid classifications diagnosed on the DSM-5.

### 3.5. Correlation between the Number of ACEs of the Parent and The Child

The young age of the children in our sample leads us to expect that the number of children’s ACEs was not high enough to establish a correlation between the number of the children’s ACEs and the number of the parents’ ACEs. This was because the latter were based on ages 0–18, and the children’s ACEs were based on ages 0–6. Although we did not expect it, a significant correlation (r = 31) was found between the number of the parental ACEs and the number of child ACEs.

### 3.6. Correlation between Risk Factors in Different Domains and the Intensity of Treatment

The mean number of interventions contributed to the family was 4.02 (SD 1.39), and the mean number of involved disciplines in these treatments was 4.74 (SD 1.49). The number of other services involved was 2.07 (SD 1.64). In a third (35%) of all the families, the integrated family approach in the treatment was finished within a year, another third (33%) was finished within two years, and almost one-third (27%) underwent a treatment duration that was longer than two years. Five families were still undergoing treatment.

We found a highly significant correlation (r = 0.60) between the total number of risk factors and the number of other services involved in addition to adult and child mental health services (see Table 9). Furthermore, a significant moderate correlation (r = 0.43) was found between the number of other services involved and the risk factors in the environmental domain, the family domain (r = 0.43), and the child domain (r = 0.40), but a significantly weak correlation was found with risk factors in the parental domain (r = 0.21). We found a significant weak correlation between the number of interventions and risk factors in the child domain (r = 0.23) and the total number of risk factors (r = 0.24). We did not find significant correlations between the risk factors in the different domains and the number of involved disciplines.

## 4. Discussion

The main objective of this study was to identify whether there was a complexity of problems and risks in a clinical sample of patients and their young children (0–6) in mental health care. Furthermore, if we identified this complexity, we aimed to determine whether there were correlations between certain risk factors, and whether this complexity was reflected in the integrated treatment given. In this sample, all parents and their infants were referred to adult and child mental health services for treatment with an integrated family approach. The latter was mostly because of concerns regarding the development of the parent–child relationship and the consequences of the parental mental disorder on parenthood. Considering that problems and risks are often mutually reinforcing, we were interested in whether these families had other risk factors in addition to the problems mentioned at the time of referral, and, if so, how many; the reason being that an accumulation of problems in addition to mental health and relational disorders has implications for treatment.

An important finding from this study was that there was an accumulation and a large variety of risk factors in the families, with a mean number of 8.43 risk factors, and that two-thirds had at least one risk factor in all of the four domains (parental, child, family, and environment). The fact that these families were not referred for complex problems makes this even more significant. These findings are in line with the expectations of our first research question, concerning how the majority of all families experience problems in the domains of the parent, the child, and the family. However, it was unexpected that the majority of them also faced challenges in their social and economic environments. Nevertheless, our findings align with the research on social and economic determinants, which have shown associations with environmental risk factors and mental disorders [15,34].

More than half of the families received support from at least two organizations in addition to the involvement of adult and child mental health services, which is also an indication of the presence of cumulative problems. This indication of a high-risk population means that most of the families in our study certainly meet the criteria of a multi-problem family [35]. In addition, it can be concluded that the families in this sample showed homogeneity with regard to the accumulation of risk factors. However, due to the broad range of risk factors, heterogeneity was observed in the combinations of risk factors in the different domains. As a result, no recommendations can be made for a standard treatment program that will be appropriate for all families. This is in line with the recommendations made by Bodden and Deković [35], regarding how interventions of multi-problem families should be multi-targeted and tailored to the characteristics of the family.

Another important finding was the high number of child traumas in comparison to an epidemiological study in the Netherlands [24], and in comparison to a youth population referred to an outpatient mental health clinic [36]. In the self-reported study [24], among children of 9–13 years of age, at least one ACE score was found in almost half of the children. Furthermore, they found that a higher number of these childhood traumas correlated to experiencing a lower quality of life. In the youth psychiatric population aged 12–18 years [36], 69.1% experienced at least one ACE, and 17.1% experienced four or more. In our sample, all children had at least one ACE score, with the mean ACE score being 2.08, and almost one in three children had at least three ACEs. Considering that far more than two-thirds (72%) of the children in our sample were younger than three years old, it highlights the necessity for timely intervention, mitigating the effects of traumas on the lives of these young children. In a recently published cohort study [37], it was concluded that positive parent–child relationships were found to be associated with better mental health and lower perceived stress in young adulthood in the context of ACEs. This confirms the significance of prioritizing the parent–child relationship in treating patients with young children, as practiced in an integrated family approach in mental health care.

Regarding the childhood traumas of the parents, four out of ten parents were themselves the child of a parent with a mental disorder. This is comparable with previous research concerning the transmission of mental disorders from one generation to the next generation [1]. Since one of the essential aims of an integrated family approach in treatment is to help parents to interrupt the cycle of the intergenerational transmission of mental disorders, we can conclude that, in 41% of this sample, there were mental health issues spanning three generations. These findings underscore the importance of helping parents to break these intergenerational cycles. Mental health treatments fall short by only treating the effects of childhood traumas in adults, without considering the risk that the next generation will be tomorrow’s patient.

There were two more findings related to the intergenerational transmission of mental health disorders. Firstly, nearly every child in this sample was exposed to maternal mental disorder and stress during pregnancy, which increases the risk of developing a difficult temperament, behavioral problems, and mental disorders [4]. Preventive approaches and active treatment were recommended to help parents to protect their infants from problems in self-regulation [38,39]. Secondly, more than half of all parents in this sample need support with child-rearing problems. Impaired caregiving, as part of family life, is assumed to be one of the important mechanisms in the intergenerational transmission of mental disorders [40,41]. These findings underscore the necessity of keeping the whole family in mind during treatment, with the focus on the individual mental health disorder of the parent, parenthood, the developing parent–child relationship, and other family members and relationships.

Regarding the second research question, we expected stronger associations between the parental, child, and family domains than between these three domains and the environmental domain. Interestingly, between the four domains, we found that the family domain was the only domain that was significantly correlated with risk factors in the other three domains. A plausible explanation can be assumed to be that family issues play a crucial role in the interaction between all risk factors related to the intergenerational transmission of mental disorders. This is in line with previous research by Kessler et al. [6], who found that childhood adversities associated with maladaptive family functioning were the strongest predictors of mental disorders over the course of the child’s lifetime, which emphasized the importance of the focus on the family in the protection of the parent and child against this intergenerational transmission of mental disorders.

Concerning our third research question concerning correlations between risk factors which are associated with the severity of the mental disorder of the parent, the data in this sample show a particular trend, namely that the higher the number of childhood traumas (ACEs), the younger the age at which the mental disorder manifested for the first time. Another related trend can be concluded in terms of the risk of the chronicity of mental health problems; the younger the age of the onset of the mental disorder, the more likely mental health problems tend to persist. Furthermore, the data indicated that the chronicity and complexity of the mental disorder are related; a correlation was found between the number of previous referrals for a mental health disorder and the number of comorbid classifications diagnosed on the DSM-5. These findings are partly in line with our expectations regarding our third research question about correlations between the four risk factors being associated with the severity of the parental mental disorder. Although we did find a correlation between childhood traumas and the age of the onset of the mental disorder, we did not discover a correlation between childhood traumas and chronicity or comorbidity.

Next, for the fourth research question about the correlation between the number of parental and child’s ACEs, we found a moderately significant positive correlation between the number of parental and child’s ACEs. Given the order in which the ACEs occurred, the direction of the correlation can only point in one direction: from parent to child. This is not in line with our expectations. Despite all of the children having at least one ACE, namely the mental disorder of their parents, we did not expect that the number of ACEs would be much higher than one because of the young age of the children in our sample (0–6 years). These findings underscore the vulnerability of these families and justify an early intervention, with the focus being on the family as a whole system.

For our fifth research question, regarding the extent to which the identified risk is reflected in the intensity of treatment, we found mixed results. Regarding the correlation between the number of risk factors and the intensity of the treatment, we found mostly significant correlations between the number of other services involved and all risk factors and the number of risk factors in each domain. These findings are in line with our expectations, namely in that, if the number of risk factors and problems was increased, more services would be involved to meet all the needs of the family. However, the expectation that the level of risk factors was also associated with the number of offered interventions from adult and child mental health services could only be confirmed for the child domain and the total number of risk factors. We did not find significant correlations between risk factors in the different domain and the total number of risk factors, or between the risk factors in different domains and the number of involved disciplines.

### 4.1. Strengths and Limitations

In this study, we have clearly demonstrated that parents with mental disorders and their young children are a high-risk population with complex problems in multiple domains. Populations with a complexity of problems are not often included in studies because of methodological problems. The results of this study were based on data obtained within a clinical setting in the Netherlands, which implies that it accurately reflects current clinical practice. Moreover, it is a reasonably realistic reflection of the educational achievements of the population in the society of the Netherlands. However, outcomes cannot simply be generalized to all clinical samples due to demographical and cultural differences. 

Another strength of this study was the involvement of families with very young children, in which the majority of the children were under three years of age, having been underrepresented in previous studies [42]. The reason for this underrepresentation is assumed to be the resistance of parents to ask for help, as well as a lack of sensitivity from professionals concerning emotional and relational problems in young children [43]. This study evidenced that infants and their parents with mental health disorders are at risk, and therefore deserve early prevention interventions and treatment. Moreover, this study demonstrated that embracing an integrated family approach in mental health care is a reasonable way to reach these infants and their parents.

An important limitation is implicit to the design of this study, a case file study, which implicated that the source of the obtained data has been recorded for treatment and not for research goals. Differences between professionals in terms of what they reported in the case file may have generated a bias. In the case of missing data, we have tried to solve this by comparing the files of the parent and the child of the same family, which contain some overlapping and some dissimilar content due to their different goals, in order to check and complete the data. Where a lack of information exists, we have checked with one of the involved professionals where possible. For this reason, we expect that, although the number of risk factors in this sample is high, it is more likely that there is an underestimation rather than an overestimation of the number of risk factors. For instance, the ACE scores of the parent and child were probably underestimated because the categories of emotional abuse and physical neglect especially are often poorly described and more implicit. In the case of the latter, we did not count it as an ACE. Because of the abovementioned limitation, we were not able to include all the risk factors mentioned in the literature as some of them were not recorded in the files, e.g., the quality of the neighborhood, belonging to a minority group, and perceived discrimination.

Another limitation is that this study did not include the protective factors in the four domains. In our literature review [14], we found that the transmission of mental disorders involves risk and protective factors. The impact of risk factors can be moderated by the presence of protective factors. To present a comprehensive picture of risk assessment in the population of parents with a mental disorder and their young children, protective factors should also be measured and considered. Consequently, an emphasis on risk factors is criticized by resilience researchers. To move away from pathological labelling, and according to positive psychology [25], attention should be focused on protective factors and strength-based intervention [44]. We subscribe to the importance of protective factors, especially in the treatment of this population.

However, there are several reasons for the focus on risk factors in this study. One of the most important is related to how the study was designed. The data source consisted of the case files of patients and their young children who were referred to mental health services due to specific problems. Professionals naturally focused on the issues for which help was sought. Another problem in the research about risk and protective factors is that many protective factors are the opposite of risk factors. For instance, a stable financial situation is mentioned as a protective factor, while financial strains, debts, and poverty are noted as risk factors. Moreover, a plain quantitative sum of risk and protective factors will not be distinctive in its degree of the risk of the actual situation. Therefore, it will not provide a realistic tool to provide insight into or reflections of the current severity of the situation. Moreover, it will not indicate which domains need to be intervened to prevent parents and their young children from the intergenerational transmission of adversity and mental disorders. It is certainly important to consider the protective factors regarding the risk factors, but expressing them both quantitatively and comparing them in terms of numbers provides no additional value. In this study, measuring protective factors in a reliable way was not possible. Protective factors which are mentioned by professionals in the files, for example, the presence of a support system, may be perceived by the parent(s) concerned to have another value. Another point to note is that some risk and protective factors are not discriminating in this sample; all families had at least one parent with a mental disorder (risk factor), and all of these parents had accepted treatment for this (protective factor). Also, almost all parents recognized that their own problems could influence (the development of) their child, which motivated them towards treatment including an integrated family approach.

### 4.2. Implications for Clinical Practice

There are a lot of implications for clinical practice to be made from this study. Firstly, due to the high number of present risk factors in the investigated families in mental health care, and the variation of these risk factors over the four domains, namely the parent, the child, the family, and the environment, it is highly recommended to conduct a broad assessment concerning risk and protective factors within the family and the environment.

Based on the interrelatedness of the risk factors, another implication can be made regarding the severity of the mental disorder of the individual adult patient (parental domain). Mental disorders manifest in the context of personal history. Since childhood traumas are related to mental health problems, these traumas should always be taken into account in mental health treatment. Whenever possible, it is preferable to treat the underlying causes of mental disorders rather than to treat only the symptoms that result from the trauma experienced in childhood.

Naturally, statistics reduce reality and make us lose sight of the individuality of the families we are talking about. It was not the goal of our study to be deterministic by showing the risks or to imply that we can predict the future by suggesting adverse outcomes for parents with mental disorders and their young children. We do not believe that problems are irreversible. We believe that, because of the complexity of our world, we have no other choice but to deal with the uncertainty of the future. Science can help us to better understand complex systems by investigating all the factors, which interact in multiple ways [45]. And, since there are so many factors that influence a family and how it functions, and since each family or parent–child relationship is unique, we will always need to provide tailored programs and find the best ‘port of entry’ (e.g., behavior, developing representations) to enable progress towards a healthier situation for each family [46]. We hope that our study will help professionals recognize the importance of helping parents to break the cycle of the intergenerational transmission of mental disorders. Therefore, families with a complexity of problems deserve multi-targeted, integrated, and tailored treatments.

Because this study focused on identifying the risk profiles of families, it could easily lead to the suggestion that the authors are only focused on risks and are thus contributing to the stigmatization of parents with mental disorders and their children. Obviously, we do not intend to be contributing to stigmatization. Although focusing on risk factors alone has its limitations, we believe it is valuable in clinical practice. Of course, in most cases, in clinical practice, it will not help to discuss the present risk factors with parents because it could lead to insecure feelings of being judged, or a loss of hope, which is an important protective factor. Recognizing the current risk factors in the four different domains can function as a compass for professionally providing guidance about what is needed to protect the parent and their children from adverse outcomes. Identifying risks in the children of parents with a mental disorder is recommended as standard practice for the adult mental health force [32,33].

Furthermore, due to the still dominant medical model, mental disorders are often viewed as an individual problem and are treated as such. The parental role of the patient, the family, including partners and children, and the socio-economic context are often ignored in treatment. The characteristics of the sample in this study clearly show that a mental disorder does not occur in isolation and is significantly correlated with problems within the family domain. We hope that this study will contribute to the awareness of the importance of making parenthood and the family part of the treatment among professionals in mental health services. Both risk and protective factors can provide guidance.

An important finding of this study is that, in 41% of the adult patients, the parents, grew up with at least one parent with a mental disorder, and thus mental health problems occurred in at least three generations. Previous research found evidence for the increased risk of the chronicity of mental disorders and poorer functioning among the children of parents with mental disorders when compared to the children of parents with no history of mental disorders [2]. Because of this, a clinical assessment of the family history of people with mental disorders is recommended to assess the long-term risks of (adult and child) patients (e.g., [1]).

Families with a complexity of interrelated problems in different domains received help from many services. More than half of the families in our sample had contact with at least four services, namely two mental health services and two other services. The mean number of involved disciplines during treatment at adult and child mental health services was almost five. This implicated that more than half of the families had contact with at least six or seven professionals. On the one hand, this can be a positive sign because it suggests that their problems were taken seriously and addressed. On the other hand, this can be a significant threat to effectiveness and efficiency [35]. In the ideal situation, the parents or caregivers were able to communicate with all of the professionals involved and take control in terms of what the family needed, at what frequency, and in which sequence. However, in families who are facing a complexity of problems and are experiencing stress, a lack of control within their lives is one of the problems. They are not able to orchestrate all the help they get. In that case, a worse scenario could be a reality in which professionals, despite their good intentions, add more stress to the family, rather than diminishing stress. It will be clear that professionals should communicate, collaborate, and prioritize their efforts, tailoring their help to the capabilities (e.g., resources, understanding, coping mechanisms) and situation of the particular family. This is one of the merits of the integrated family approach in mental health care.

In our previous study, parents told us that the communication between professionals, the multidisciplinary consultation, was one of the successful factors of the integrated family approach in their treatment [47]. Also, professionals could be overwhelmed when facing a complexity of problems in families. Like parents, they mentioned the multidisciplinary consultations as one of the key elements of success in an integrated family approach within treatment [30]. It provides them with multiple perspectives of the whole family and clarifies the complexity. It enables them to feel comfortable when coping with the complex needs of the family. The process of reflection and mentalization within the multidisciplinary consultation regulates them and provides a sense of control. An integrated family approach within treatment enables professionals to keep the whole family in mind and tailor their treatments to each other, allowing it to become a coherent effort.

## 5. Conclusions

The study of this sample shows clearly that mental disorders among most of the patients, who were all parents of young children, was only one of the problems they have to deal with. The cumulation and interrelatedness of these problems increased the risk of the intergenerational transmission of mental disorders. To prevent these parents and their young children from being caught up in this intergenerational cycle, malleable risk factors should be addressed in treatment. Thus, an integrated family approach to treatment is recommended as a way of achieving this.

## Figures and Tables

**Figure 1 ijerph-21-00640-f001:**
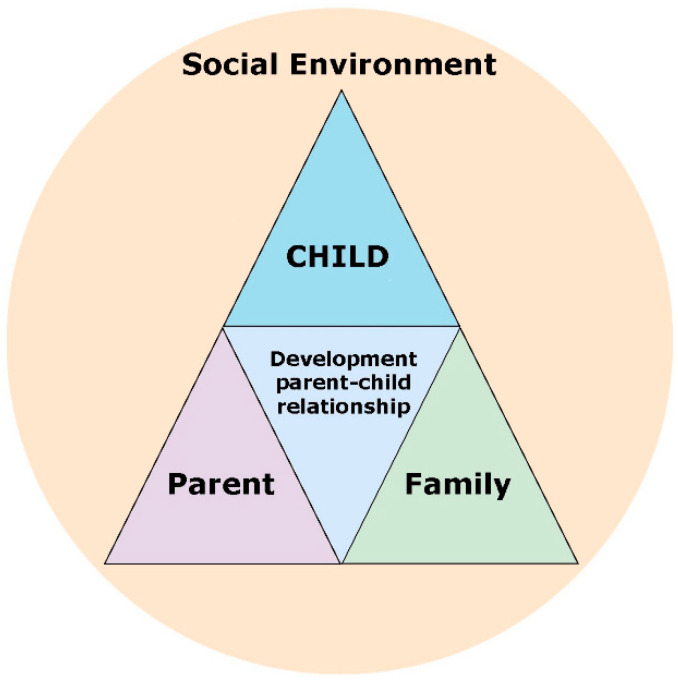
Interrelated domains affecting the developing parent–child relationship.

**Table 1 ijerph-21-00640-t001:** Characteristics of the parents and their children (N = 100).

**Adults/Parents**		N	**Children**		N
Gender	Woman	85	Gender	Girls	52
	Man	15		Boys	48
Age (years)	<20	1	Age (months)	0–12	46
	20–30	36		12–24	16
	>30	63		24–36	10
				36–48	9
				48–60	10
				60–72	9
**Highest Educational Level of the Parent by Referral**					N
Low (basic or pre-vocational secondary education)					20
Middle (secondary vocational education)					50
High (bachelor’s or master’s degree)					30

**Table 2 ijerph-21-00640-t002:** The frequency and percentages of the presence of 26 risk factors in different domains in all 100 families.

	N	%
**Risk factors among the parents with a mental health disorder**	100	100
Early onset of the mental disorder (before age 30)	90	90
≥2 previous mental referrals to health care services (chronicity)	56	56
Comorbidity of DSM-5 classifications	70	70
≥4 ACE score	37	37
Number of parents with at least one risk factor in the parental domain	97	97
**Risks among the child domain**	100	100
Anxiety/panic/stress during pregnancy *	70	82
Previous referral to youth or mental health care (chronicity)	7	7
Comorbidity (severity)	22	22
≥3 ACEs	29	29
Number of children with at least one risk factor in the child domain	81	81
**Risks among the family domain**	100	100
Single parent	21	21
Early parenthood (age below 20)	11	11
Both parents have mental health problems	8	8
Family member is addicted	18	18
Parents divorced/split up	29	29
Problems in couples’ relationship	13	13
Domestic violence	18	18
Other child with problems	8	8
Life events during treatment	39	39
Low education attainment	20	20
Child-rearing problems or unpredictable or lack of daily routines	56	56
Number of families with at least one risk factor in the family domain	91	91
**Environmental risks**	100	100
Unemployment/problems at work	67	67
Housing problems	15	15
Financial strains/debts/poverty	33	33
≥2 other services involved	57	57
Absence social support network	7	7
Criminality	3	3
Problems within the extended family	37	37
Number of families with at least one risk factor in the environmental domain	87	87

* N = 85 mothers.

**Table 3 ijerph-21-00640-t003:** Mean of risk factors in all four domains and of all 26 risk factors.

	N	M	SD	Range
Parental domain	100	2.53	1.02	0–4
Child domain	100	1.30	0.89	0–4
Family domain	100	2.41	1.56	0–7
Environmental domain	100	2.19	1.48	0–5
All 26 risk factors	100	8.43	3.16	1–16
Risks in all 4 domains	100	3.56	0.68	1–4

**Table 4 ijerph-21-00640-t004:** Mean of risk factors which have to do with the severity of the mental disorder of the adult patients (parental domain).

Parental Risk Factors	N	M	SD	Range
Age of first onset of a mental disorder during lifetime	80	20.9	7.71	3–37
Number of previous referrals to mental health care services (chronicity)	93	3.13	1.77	1–9
Number of DSM-5 classifications	100	2.17	1.06	1–5
Number ACE score	100	2.84	2.16	0–8

**Table 5 ijerph-21-00640-t005:** Frequency of the adverse child experiences of parent and child.

Adverse Child Experiences		Parent Age 0–18 N = 100	Child Age 0–6 N = 100
Abuse	Physical abuse	29	8
	Emotional Abuse	26	8
	Sexual Abuse	26	1
Neglect	Physical neglect	10	1
	Emotional neglect	71	26
Household disfunction	Parent with mental disorder	41	100
	Violence between parents	16	14
	Growing up without both biological parents	45	29
	Substance abuse within the family	18	18
	Incarcerated member of the family	2	4

**Table 6 ijerph-21-00640-t006:** Number of parental and child adverse childhood experiences.

Number of ACE Score	N	M	SD	Range
Parent	100	2.84	2.16	0–8
Child	100	2.08	1.43	1–8

**Table 7 ijerph-21-00640-t007:** Correlations between the parental, child, family, and environmental domain.

Domains		Parent	Child	Family	Environ-Ment
Parental domain	Pearson Correlation	1	0.267 **	0.312 **	0.000
	Sig. (2-tailed)		0.007	0.002	0.996
	N	100	100	100	100
Child domain	Pearson Correlation	0.267 **	1	0.244 *	0.117
	Sig. (2-tailed)	0.007		0.015	0.245
	N	100	100	100	100
Family domain	Pearson Correlation	0.312 **	0.244 *	1	0.229 *
	Sig. (2-tailed)	0.002	0.015		0.022
	N	100	100	100	100
Environmental domain	Pearson Correlation	0.000	0.117	0.229 *	1
	Sig. (2-tailed)	0.996	0.245	0.022	
	N	100	100	100	100

** Correlation is significant at the 0.01 level (2-tailed); * Correlation is significant at the 0.05 level (2-tailed).

**Table 8 ijerph-21-00640-t008:** Correlations between the four risk factors of the adult patient (parent).

Parental Risk Factors		Age of Onset	Comorbidity	Chronicity	ACE Score
Age at the onset of mental disorder	Pearson Correlation	1	−0.202	−0.310 **	−0.364 **
	Sig0. (2-tailed)		0.071	0.006	0.001
	N	81	81	78	81
Comorbidity: number of DSM-classifications	Pearson Correlation	−0.202	1	0.324 **	0.091
	Sig0. (2-tailed)	0. 071		0.002	0.367
	N	81	100	93	100
Chronicity: number of referrals to mental health care services	Pearson Correlation	−0.310 **	0.324 **	1	0.154
	Sig0. (2-tailed)	0.006	0.002		0.141
	N	78	93	93	93
ACE score	Pearson Correlation	−0.364 **	0.091	0.154	1
	Sig0. (2-tailed)	0.001	0.367	0.141	
	N	81	100	93	100

** Correlation is significant at the 0.01 level (2-tailed).

**Table 9 ijerph-21-00640-t009:** Correlations between risk factors in domains and the number of interventions, disciplines, and other services involved.

	Intensity Treatment	Number of Interventions	Number of Involved Disciplines	Number of Other Services Involved
Risk Factors in Domains				
Parental domain	Pearson Correlation	0.163	0.130	0.213 *
	Sig. (2-tailed)	0.105	0.196	0.033
	N	100	100	100
Child domain	Pearson Correlation	0.231 *	0.149	0.399 **
	Sig. (2-tailed)	0.021	0.139	0.000
	N	100	100	100
Family domain	Pearson Correlation	0.094	0.125	0.433 **
	Sig. (2-tailed)	0.354	0.214	0.000
	N	100	100	100
Environmental domain	Pearson Correlation	0.170	−0.048	0.429 **
	Sig. (2-tailed)	0.090	0.633	0.000
	N	100	100	100
Total number of risk factors in all domains	Pearson Correlation	0.244 *	0.124	0.596 **
	Sig. (2-tailed)	0.015	0.220	0.000
	N	100	100	100

** Correlation is significant at the 0.01 level (2-tailed); * Correlation is significant at the 0.05 level (2-tailed).

## Data Availability

The raw data will be available on request.

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
