# Peer review of "The Integrated Family Approach in Mental Health Care Services: A Study of Risk Factors"

_ijerph, 2024, doi:10.3390/ijerph21050640_

Round 1
Reviewer 1 Report
Comments and Suggestions for Authors
Dear authors,
I read your article with great interest and I consider it important from the point of view of the problem addressed.
The primary areas of focus for the research of the risk factors in mental health care are the transmission of mental diseases across generations and the management strategies for this extremely complicated phenomenon.
For an easier reading of the important elements of the article, I suggest that you attach some of the tables as additional data.
Although the manuscript is interesting and the statistical analysis performed is correct, the results obtained are already known. For an additional scientific contribution, I think it would be interesting to build a regression model with predictive value, taking into account these risk factors that you have already analyzed in the article.
Finally, I consider your manuscript worthy of publication after major revision.
Author Response
Dear reviewer,
Please, find our response in attached file.
The authors

Reviewer 2 Report
Comments and Suggestions for Authors
The introduction provides a thorough overview of the literature concerning intergenerational transmission of mental disorders and associated risk factors. The manuscript outlines the importance of early childhood in mental health development and highlights the significance of parental, child, family, and environmental factors. The research questions are clearly stated, providing a roadmap for the study. However, there are a few minor issues to address such as the consistency in citation style throughout the introduction. Clarify the exact methodology and objectives of the study to enhance understanding for readers. Integrating recent literature to strengthen the theoretical framework would be beneficial.
There are a few minor corrections and suggestions to enhance clarity of the Materials and Methods section. In the Participants and Recruitment subsection, clarify whether the treatment was ongoing or completed at the time of recruitment for families in the two groups. In the Procedure, Data Collection subsection, specify the training and qualifications of the researchers who collected and analyzed the data. In the Measurements subsection, provide a brief rationale for the selection of each risk factor to strengthen the justification for their inclusion. Consider organizing the risk factors under subheadings (e.g., individual, child, family, environmental) for better readability. In the Data Analysis subsection, provide a brief rationale for choosing Pearson's correlation coefficient for analyzing correlations between variables. Clarify any assumptions made when using Pearson's correlation coefficient, especially regarding the normality of the data distribution.
The authors need to strengthen the results section. In the sentence "Far more than half of all families had between 6-10 risk factors (56%), followed by a quarter of the families with 11-17 risk factors (25%)," it would be clearer to specify the total number of families considered in the study. The phrase "two thirds of the families" should be written as "two-thirds of the families" for consistency. In the sentence "In more than one third of the families (39%), there was a life event during treatment such as physical illness of a family member, birth of a child, or relocation," consider specifying the total number of families surveyed. The phrase "one third of the families" should be consistent throughout the text, either as "one-third of the families" or "one third of the families." In the sentence "Regarding risk factors related to social support: in 37 families there were problems with the extended family and in seven families there was no social support," consider specifying the total number of families surveyed. In Table 2, "Risk factors among the parents with a mental health disorder" should have "N = 100" rather than "N = 97" to match the total number of families. In Table 4, "Parental risk factors" should include the total number of families surveyed to provide context. In Table 9, "Adverse Childhood Experiences (ACEs)" should specify the total number of parents and children surveyed. In Table 10, consider providing the total number of families surveyed to offer context for the treatment characteristics. In Table 11, specify the total number of families surveyed for each correlation analysis.
The discussion section needs improvement. Line 450: "Moreover, if we found this complexity" - It seems there may be a missing word or a typo here. It could be clarified to something like "Furthermore, if we identified this complexity..." Line 558: "We did not found significant correlations" - It should be "We did not find significant correlations." Line 655: "port of entry" - It seems there might be a typo here. Did you mean "point of entry"? Line 672: "The parental role of the patient, the family with partner and children, and the social economic context is often ignored in treatment." - It might be clearer if rephrased as "The parental role of the patient, the family including partners and children, and the socio-economic context are often ignored in Line 702: "professionals should communicate, collaborate, and prioritize their efforts, tailoring their help to the capabilities and situation of the particular family." - It might be clearer to specify what "capabilities" refer to in this context. For example, "tailoring their help to the capabilities (e.g., resources, understanding, coping mechanisms) and situation of the particular family." Line 710: "It enables them to feel comfortable to cope with this complexity." - It might be clearer to specify what "this complexity" refers to. For example, "It enables them to feel comfortable coping with the complex needs of the family." These are just minor suggestions to enhance the clarity and precision of the discussion section. Overall, the discussion provides a thorough analysis and valuable insights into the study findings and their implications.
Comments on the Quality of English Language
Minor editing is required.
Author Response
Dear reviewer,
Please, find our response to your comments in attached file.
Kind regards,
The authors

Reviewer 3 Report
Comments and Suggestions for Authors
The Integrated Family Approach in Mental Health Care Services: A Study of Risk Factors
Review for ijerph
This is an interesting and promising study, but further analysis is needed to refine the experimental control.
In the introduction, the authors argue that exposure to stress early in life is associated with later psychopathology. The authors draw on two studies, Agorastos et al., 2019 and Van den Bergh et al., 2017. Could the authors go a little further in their explanations? What types of psychopathology exactly?
The same is true when the authors say that, according to Barker et al. (2012), in the presence of maternal depression at age 2, each additional risk factor increases the risk of mental disorder in the child at age 7 by at least 20%. Which mental disorder? It's a very broad generic term used to describe dozens of pathologies. Can the authors refine their statement? There's a world of difference between childhood depression and childhood psychosis...
According to APA standards, the titles of figures must be below these. Check the journal's criteria.
It seems problematic to consider the following mental disorders as a whole: personality disorder, bipolar disorder, depressive disorder, anxiety disorder¬, der, autism spectrum disorder, post-traumatic disorder and other specified traumas, and stress-related disorders. Unless I'm mistaken, the authors didn't correlate disorder type with disorder type, yet risk factors vary from one disorder to another, as do effects on parenting. Further analyses should certainly be provided.
The authors mention that, with regard to parental childhood trauma, four out of ten parents were themselves the child of a parent with a mental disorder. Here again, no details of the disorder are given. Is it the same disorder as the parent's? A different disorder? The authors speak of the transmission of mental disorders from one generation to the next, but without specifying whether the parent's disorder was the same as the child's... Not all torubles are comparable in terms of symptoms, symptom intensity, impact on social, personal and professional life, and so on...
Author Response
Dear reviewer,
Please, find our response to your comments in attached file.
With kind regards,
The authors

Round 2
Reviewer 1 Report
Comments and Suggestions for Authors
Dear authors,
I consider that your manuscript has been improved and can be published.
Kind regards.
Author Response
Dear Reviewer,
Thank you for your positive conclusion that our manuscript is improved en in your opinion can be published.
With kind regards,
The authors.
Reviewer 3 Report
Comments and Suggestions for Authors
Experimental control is essential in scientific research to mitigate the effects of unwanted variables and thereby ensure the validity of results. When a study exhibits a high level of experimental noise, it means that various uncontrolled factors may influence the outcomes, thus diminishing the ability to detect significant effects. Consistent with Cohen's work, high experimental noise can result in small effect sizes, complicating data interpretation and potentially leading to less reliable conclusions. Therefore, careful attention must be paid to minimizing experimental noise, whether through rigorous control techniques, experimental design, or appropriate statistical analysis, to ensure the robustness of the findings
The problem is that if you don't specify the group or groups studied, you add noise. In the end, the effects observed may be false positives, in the sense that many spurious variables are not controlled.
Author Response
Our reply to reviewer 3, round 2.
Dear reviewer,
Thank you for explaining the possibility of experimental noise threatening the validity and reliability of research with an experimental design. We agree that it is important to make every effort to minimize this. In this part of our entire study, we did not evaluate the effect of an intervention, but instead examined a clinical sample about the presence of risk factors, other than the reason of referral. The clinical sample consisted of patients and their young children who received mental health treatment with an integrated family approach. In another paper, which is under review, we evaluate the effect of the treatment. The aim of the study described in this paper was to understand whether there was a complexity of problems in this sample and whether we could find associations between certain risk factors. Furthermore, we were also interested in whether these risk factors were addressed in the treatment given.
We hope that we have answered your questions satisfactorally.
With kind regards,
The authors